# Automatic Source Code Summarization via Reinforcement Learning

**Zhuangbin Chen**
Department of Computer Science
The Chinese University of Hong Kong
Shatin, Hong Kong
1155100476@link.cuhk.edu.hk

**Yuelong Shen**
Department of Computer Science
The Chinese University of Hong Kong
Shatin, Hong Kong
1155156114@link.cuhk.edu.hk

## Abstract

Due to the fast development of computer and software, the volume of today's code has reached an unprecedented level. For large-scale systems (e.g., cloud computing systems) with billions lines of codes, the majority of its maintenance effort is code management. And much of this effort is spent on understanding related source codes. With high-quality code summaries, one can quickly understand what a function does (even without reading the code). However, it's nontrivial for a programmer to write good comments for source codes. If the code summary can be automatically generated, then we can greatly accelerate the whole pipeline of software development. To this end, in this project we develop a reinforcement learning framework to enhance the automatic generation of code summarization. By properly defining the key modules, we employ an actor-critic network to solve this problem. Experimental results demonstrate that our model outperforms vallina sequence to sequence model by a noticeable margin. Introduction video of this project is available here[1].

## 1   Introduction

Code summarization is to provide a compact and informative natural language description for the functionality of a code snippet. Such summaries can not only facilitate the understanding of programmers, but also benefit the task of software maintenance. Figure 1 illustrate an example of Python code snippet together with a comment that summarizes the high-level function of the code. With high-quality code summaries, one can quickly understand what a function does (even without reading the code). However, writing summary could be a massive burden for developers, hence only a small fraction of all code will be paired with a comment. Therefore, it is of practical importance to automate this process, as done by some existing work [1, 2, 3, 4].

Reinforcement learning (RL) is one of the basic machine learning paradigms, which is able to interact with an environment and learn what actions to take in order to maximize the cumulative reward. Due to the exceptional ability of learning without labels, it has been employed in many applications, such as robotics, video games, natural language processing, and computer vision.

In this proposal, we plan to apply the principles of reinforcement learning to help the process of automatic generation of code summaries. Specifically, we follow the popular setting of data-driven code summarization which is composed of an encoder (learn the latent representation of code snippets) and a decoder (generate code comments based on the representation). In the decoding phrase, we leverage reinforcement learning to assist the selection of output words that can achieve a better performance.

---

[1]introduction video

```python
"""Sets the seed for generating random numbers to a random number on all GPUs. It's
    ↪ safe to call this function if CUDA is not available; in that case, it is
    ↪ silently ignored."""
def seed_all() -> None:
    random_seed = 0
    seeded = False
    for i in range(device_count()):
        default_generator = torch.cuda.default_generators[i]
        if not seeded:
            default_generator.seed()
            random_seed = default_generator.initial_seed()
            seeded = True
        else:
            default_generator.manual_seed(random_seed)
```

Figure 1: A Python code snippet excerpted from Pytorch, where the comment above the function is the corresponding code summary.

The remainder of this report is organized as follows. Section 2 briefly introduces some related work about code summarization. Section 3 presents some preliminaries of language model and neural sequence generation model. Section 4 elaborates on our reinforcement learning framework for code summarization. Section 5 shows the experimental results. Finally, Section 6 concludes this project.

## 2   Related Work

In the literature, some effort has been devoted to the automatic generation of code summary. For example, [1, 2] attempted to generate comments for code based on templates or rules, which was done by learning the semantic representation of code using statistical language models. Sridhara et al. constructed a database containing a software word usage model. Then, they defined a series of rules about tokenized function/variable names, which are used to generate comment. Other statistical models include topic models and n-grams, for example, Movshovitz-Attias et al. predict comments from Java source files.

Recent work [3, 4] pursues this goal by resorting to more advanced models, e.g., deep neural networks, which mainly follow an encoder-decoder framework. Specifically, they formulate code summarization as a neural translation problem and employ Recurrent Neural Networks (RNN e.g., LSTM) to encode the code snippets and utilize another RNN to decode that hidden state to coherent sentences.

Particularly, there are also some attempts to improve the quality of code summarization using reinforcement learning, such as [5, 6]. In this project, we intend to have a deep understanding of how reinforcement learning can facilitate the auto-generation of code summaries for code snippets. More importantly, we will explore possible ways towards achieving a better performance.

## 3   Preliminaries

### 3.1   Notations and Terminologies

Given a (code snippet, code summary) pair, we first tokenize them by the symbol *space* and then remove meaningless stop words like "a", "an", and "the". Let $\mathbf{x} = (x_1, x_2, ..., x_{|\mathbf{x}|})$ denote a word sequence of source code snippet, $\mathbf{y} = (y_1, y_2, ..., y_{|\mathbf{y}|})$ denote a word sequence of code summary, where $|\cdot|$ is the length of sequence. Particularly, we use $y_{i:j}$ to denote $(y_i, y_{i+1}, ..., y_j)$. Our goal is to correctly generate $\mathbf{y}$ given the corresponding $\mathbf{x}$. $\mathcal{D} = (\mathbf{x}_1, \mathbf{y}_1), (\mathbf{x}_2, \mathbf{y}_2), ..., (\mathbf{x}_N, \mathbf{y}_N)$ is the training set, where $N$ denotes the number of training instances.

## 3.2 Language Model and Semantic Representation for Code

Language model computes the probability of occurrence of a word sequence. We denote the probability of a sequence of $N$ words $\{x_1, x_2, ..., x_N\}$ as $p(x_1, x_2, ..., x_N)$, which is formally defined as:

$$p(x_{1:N}) = \prod_{i=1}^{N} p(x_i | x_{1:i-1}) \tag{1}$$

In our project, we adopt neural language model as the language model, which reads the words in the sentence one by one, and predicts the next word at each time step. At time step $t$, the probability of the next word $p(x_{t+1}|x_{1:t})$ is estimated as:

$$\begin{aligned} p(x_{t+1}|x_{1:t}) &= g(\mathbf{h}_t) \\ \mathbf{h}_t &= f(\mathbf{h}_{t-1}, e(x_t)) \end{aligned} \tag{2}$$

where $e$ is the embedding layer that transforms input word $x_t$ into a continuous vector space, $\mathbf{h}_t$ is the hidden state at time $t$, which is generated based on previous hidden state $\mathbf{h}_{t-1}$ and the current input $x_t$, g is a stochastic output layer (typically a softmax for discrete outputs) that generates output tokens. Particularly, the RNN that computes the hidden state $\mathbf{h}_t$ is called encoder, and decoder calculates $p(x_{t+1}|x_t)$. The last hidden state of the encoder is often regarded as the semantic representation of a code snippet. Finally, the objective of the language model is given as follows:

$$\max_{\theta} \mathcal{L}(\theta) = \max_{\theta} \mathbb{E}_{(x,y) \sim \mathcal{D}} \log p(\mathbf{y}|\mathbf{x}; \theta) \tag{3}$$

where $\theta$ represents model parameters. LSTM (Long Short-Term Memory) is adopted in this project due to its exceptional ability to model the complex relationship of sequential data. The LSTM calculates the hidden state $\mathbf{h}_i$ as follows:

$$\begin{aligned} i &= \sigma(\mathbf{W}_i \mathbf{h}_{t-1} + \mathbf{U}_i x_t + b_i) \\ f &= \sigma(\mathbf{W}_f \mathbf{h}_{t-1} + \mathbf{U}_f x_t + b_f) \\ o &= \sigma(\mathbf{W}_o \mathbf{h}_{t-1} + \mathbf{U}_o x_t + b_o) \\ g &= tanh(\mathbf{W}_g \mathbf{h}_{t-1} + \mathbf{U}_g x_t + b_g) \\ g_t &= f \odot g_{t-1} + i \odot g \\ \mathbf{h}_t &= o \odot tanh(g_t) \end{aligned} \tag{4}$$

where $\sigma$ is the element-wise sigmoid function and $\odot$ is the element-wise product; $\mathbf{U}_i, \mathbf{U}_f, \mathbf{U}_g, \mathbf{U}_o$ denote the weight matrices of different gates for input $x_t$ and $\mathbf{W}_i, \mathbf{W}_f, \mathbf{W}_g, \mathbf{W}_o$ are the weight matrices for hidden state $\mathbf{h}_t$; while $b_i, b_f, b_g, b_o$ represent the bias vectors.

## 4 Methodology

One important problem of previous approaches is that they do not directly optimize the goal. Specifically, when code summaries are generated, some metrics will be applied to calculate the performance. Such metrics are usually different from the loss defined by the network. Moreover, previous approaches generate words at each time step, which lack a global view to guide the learning of the model. Reinforcement learning is an ideal solution to address this problems. This is because it allows us to directly set the evaluation metric as the reward. And we can calculate the reward for a code summary when it is completely generated.

### 4.1 Reinforcement Learning

We formulate the task of code summarization as a reinforcement learning problem by defining the key modules, in which an agent interacts with the environment in discrete time steps. At

each time step $t = 1, 2, ..., T$, the agent produces an action, i.e., a word, sampled from the **policy** $\pi(\widehat{\mathbf{y}}|\mathbf{x}; \theta) = p(\widehat{\mathbf{y}}|\mathbf{x}; \theta)$. After that, the agent transits to the next time step $t + 1$ and reaches a new **state** $\mathbf{s}_{t+1} = (\widehat{\mathbf{y}}_{1:t}, \mathbf{x}, \mathbf{y})$. Finally, the **reward** $r(\widehat{\mathbf{y}}_{1:T}, \mathbf{y})$ can be calculated when the agent travels through the entire code snippet sequence and produces a code summary sequence $\widehat{\mathbf{y}}_{1:T}$. This is related to the evaluation metric and we adopt a widely-used one called BLEU [7], which uses a modified form of precision to compare a predicted sequence against the ground truth sequence. Therefore, the accumulative reward is the gain of BLEU.

In this project, we adopt a deep reinforcement learning framework named Actor-Critic network to assist decoder network in generating code summaries based on the semantic representation of code snippets.

### 4.1.1 Actor Network

We let actor network take the task of code summary generation, which is essentially the encoder network. We incorporate attention mechanism into our model, whose effectiveness has been demonstrated in a variety of neural machine translation tasks. Specifically, at the $t^{th}$ step of the decoding process, the attention score for the $i^{th}$ token of the code summary can be calculated as follows:

$$\alpha_t(i) = \frac{exp(\mathbf{h}_i \cdot \mathbf{s}_t)}{\sum_{k=1}^{n} exp(\mathbf{h}_k \cdot \mathbf{s}_t)} \tag{5}$$

where $n$ is the number of tokens in the code snippet; $\mathbf{h}_i \cdot \mathbf{s}_t$ is the inner project of $\mathbf{h}_i$ and $\mathbf{s}_t$, which directly measures their similarity. Finally, the $t^{th}$ context vector $\mathbf{c}_t$ is calculated by summing over the hidden states of different input tokens with weight $\alpha_t(i)$:

$$\mathbf{c}_t = \sum_{t=1}^{n} \alpha_t(i) \cdot \mathbf{h}_i \tag{6}$$

Finally, we add an additional hidden layer to utilize $\mathbf{c}_i$ and $\mathbf{s}_t$ simultaneously:

$$\mathbf{c}'_t = tanh(\mathbf{W}_c \cdot [\mathbf{s}_t; \mathbf{c}_t] + b_c) \tag{7}$$

where $[\cdot; \cdot]$ denotes the concatenation operation. Let $p_\pi$ denote a policy $\pi$ described by the actor network, $p_\pi(y_t|\mathbf{s}_t)$ denote the probability of generating $t^{th}$ word $y_t$, we can have the following equation:

$$p_\pi(y_t|\mathbf{s}_t) = softmax(\mathbf{W}_s \cdot \mathbf{c}'_t + b_s) \tag{8}$$

### 4.1.2 Critic Network

Traditional encoder-decoder framework generates sequence by maximizing the likelihood of next word given previous generated words. Such method is not directly working on the ultimate goal of text generation tasks, i.e., BLEU. This is where reinforcement learning can play a role. We apply a critic network to approximate the value of generated action at time step $t$. We first introduce the value function. Given the policy $\pi$, sampled actions actions (i.e., probable words), and reward function, the value function $v^\pi$ is defined as the prediction of total reward $\mathbf{s}_t$ at step $t$, which can be calculated as:

$$v^\pi(\mathbf{s}_t) = \mathbb{E}_{s_{t+1:T}, y_{t:T}} [\sum_{l=0}^{T-t} r_{t+l}|y_{t+1}, ..., y_T, \mathbf{h}] \tag{9}$$

where $T$ is the maximum step of decoding; $\mathbf{h}$ is the representation of code snippet. Particularly, we can only obtain the final reward (BLEU) when the process of sequence generation terminates; otherwise, the reward will be 0. Finally, the objective of the critic network is as follows:

$$J(\psi) = \frac{1}{2}\|v^\pi(\mathbf{s}_t) - v^\pi_\psi(\mathbf{s}_t)\|^2 \tag{10}$$

where $v^\pi(\mathbf{s}_t)$ is the target value, $v_\psi^\pi(\mathbf{s}_t)$ is the value predicted by critic network, and $\psi$ is critic network's parameters.

## 5 Experiments

In this section, we first introduce the dataset used in our experiments and then show the experimental results.

### 5.1 Dataset

We use the dataset collected by Barone et al. [8], which is obtained from GitHub. The dataset contains 108,726 code-comment pairs, in which the comment is regarded as a summary for the code. We follow the preprocessing steps in [8]. Moreover, we shuffle the dataset and use the first 60% for training, 20% for validation and the remaining for testing. The vocabulary size of the training data is 50004, while the testing data has 31227 word

### 5.2 Evaluation Metrics

To evaluate the performance of the proposed method, we employ two evaluation metrics: BLEU [7] and perplexity [2].

BLEU measures the similarity between the candidate and the reference. In our experiments, the code summary generated by our method is regarded as the candidate, while the original code comment written by the developer is regarded as the reference. Specifically, BLEU is calculated as follows:

$$BLEU = BP \cdot \exp(\sum_{n=1}^{N} w_n \log p_n),$$ (11)

where $BP$ is a brevity penalty that penalizes overly short candidates; $N$ is the maximum number of grams used in the experiments; $p_n$ is the modified n-gram precision; and $w_n$ is the weight of each $p_n$.

Perplexity gauges how well a probability distribution or probability model predicts a sample. A low perplexity indicates the probability distribution is good at predicting the sample. Particularly, the perplexity of a discrete probability distribution $p$ is defined as:

$$perplexity(p) = 2^{H(p) = 2^{-\sum_x p(x) log_2 p(x)}}$$ (12)

where $H(p)$ is the entropy of the distribution and x ranges over events.

### 5.3 Vallina Sequence to Sequence Model

Our baseline model is a vallina sequence to sequence model [9], which predicts words by following the principle of maximum likelihood estimation. In our experiments, we feed a code snippet to it and the goal is to make its output as similar as the ground truth. The training loss is shown in Figure 2. We can see the training loss indeed decreases with iteration. Particularly, it first drops dramatically at the first a few iterations. Then, it goes through a gentle decreasing period. Finally, the loss converges and we stop the training. We also report the corpus reward (the average reward over the testing data) and perplexity of the vallina seq2seq model, as shown in Figure 3. From Figure 3(a), we can see the reward (BLEU) increases, which indicates that vallina seq2seq model is capable of learning the ground truth comments. However, the absolute value of the reward is too small, i.e., 8, which is far from satisfactory. Interestingly, the perplexity increases (recall that the smaller a perplexity is, the better) with the training epoch. We believe the reason is two-fold: 1) the perplexity is not directly the optimization goal, so the training direction deviates from this metric; 2) our vallina seq2seq is not expressive enough to learn a good function to fit the distribution. Due to the limitation of computing sources, we did not use a very large network. More exploration will be conducted in our future work.

---

[2]https://en.wikipedia.org/wiki/Perplexity

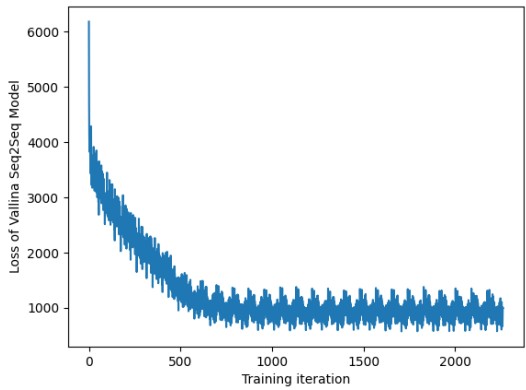

Figure 2: The training loss of vallina seq2seq model

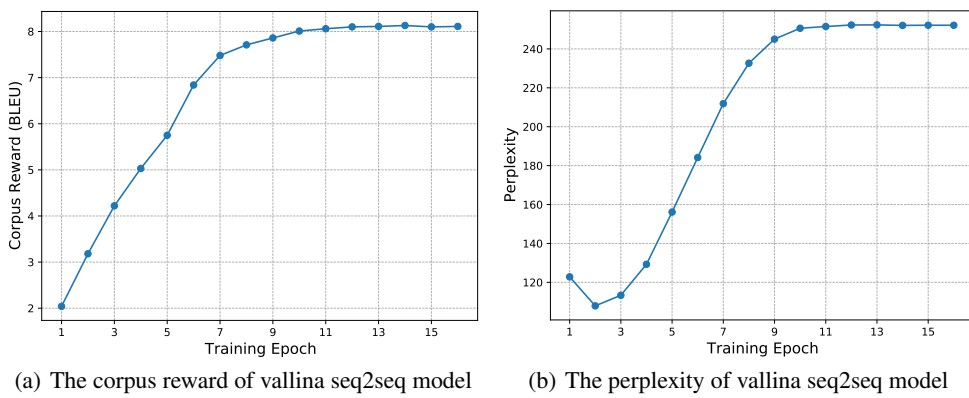

(a) The corpus reward of vallina seq2seq model
(b) The perplexity of vallina seq2seq model

Figure 3: The performance of vallina seq2seq model

## 5.4 Actor-Critic Network

The experimental results of our actor-critic network for improving code summarization are shown in Figure 4. Similarly, we report the corpus reward and perplexity. From Figure 3, we can see that the vallina seq2seq reaches its performance limit at the $9^{th}$ training epoch. Therefore, we start the reinforcement learning right from this point. In both evaluation metrics, a noticeable performance gain is achieved. For corpus reward, the best value is now more than 14, meaning a 50% performance improvement. For perplexity, right after applying reinforcement learning, its value starts to drop. This means our actor-critic network is able to guide the model to the right direction. However, the performance of our method still cannot meet our expectation. According to our analysis, the dataset collected by Barone et al. [8] contains much noise. Specifically, some comments do not precisely reflect the functionalities of the corresponding code snippet. Also, many comments are too short. For future work, we plan to collect high-quality code-comment pairs, over which we believe a much better performance can be made. Nevertheless, from our experiments, we still see our method shedding lights on applying reinforcement learning to automated code summarization.

## 6 Conclusion

In this project, we utilize a popular reinforcement learning framework, i.e., actor-critic network, to automate the generation of code summary. Specifically, we treat each time step of the decoder-encoder network as a state and each word from the vocabulary as an action. Our actor-critic network then tries to learn a policy which can improve the BLEU score of the generated code summary. Comprehensive experiments on a real-world dataset from GitHub show that our model outperforms the vallina sequence to sequence model in terms of both corpus reward (BLEU) and perplexity. Knowing that the performance achieved is still not satisfactory, we plan to collect more high-quality dataset in

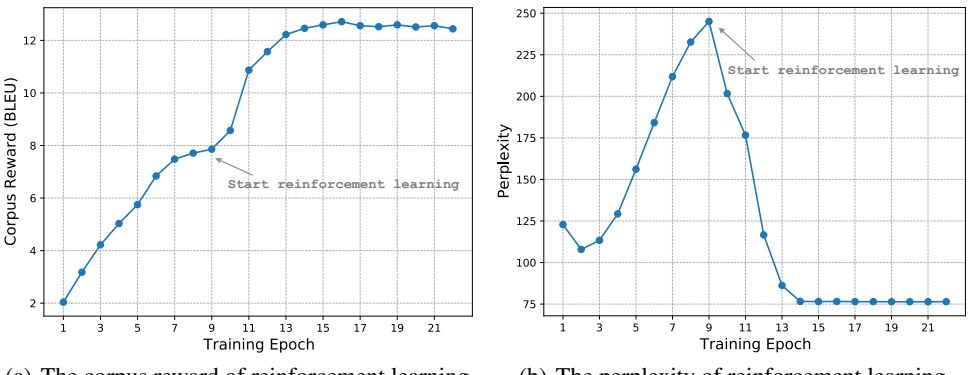

(a) The corpus reward of reinforcement learning  (b) The perplexity of reinforcement learning

Figure 4: The performance of reinforcement learning

future work. Meanwhile, more sophisticated deep learning mechanisms such as attention and pointer can be considered to further boost the performance.

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
