# OpenReview forum: "Automatic Source Code Summarization via Reinforcement Learning"
_CUHK.edu.hk/2021/Course/IERG5350_

### Official Review · AnonReviewer1 · 2020-12-18
**Well organized paper with great model description and results demonstration**

**Rating:** 8
**Confidence:** 5

**Review:**

The author proposes to apply the RL to help the automatic code summaries.  The paper is well written and organized.

1. the model is clearly described and can improve the performance on the code summary task significantly.
2. Since the DL algorithms can choose the evaluation matrics (BLEU) as the loss function directly, it seems that the RL model can easily learn this trick and get high value when testing. How about the performance with respect to other indexes, Cider?
3. How to deal with the intra-variance among the comments. The code may have different comments caused by different writing styles. The word overlapping constraints may be too strict to guide the learning process.

---

### Official Review · AnonReviewer3 · 2020-12-19
**Valuable and outstanding attempt to optimize Source Code Summarization**

**Rating:** 7
**Confidence:** 3

**Review:**

I am not very familiar with the relavant work in the language processing field, but the model present in this paper is somehow easy to follow and I will make my review based on the report and the knowledge I have.

**Significance**: This paper explored the potentiality of utilizing reinforcement learning in code summarization. Though there are existing works focusing on auto code summarization, the work in the paper points out the drawback of the previous methods and using A2C network to improve the Vallina Sequence to Sequence Model. The result showed that the A2C network can achieve significant improvement on BLEU reward and perplexity.

**Technical Quality**: Good and clear with detailed formula and definition for the environment and the model.

**Clarity**: It is easy to follow the model and the environment used in this project.

**Comments**:
1. A question is that why using A2C network on the results of the trained the Vallina model? (For comparison?)
2.  Comparisons with other RL methods in relavant work are missing.
3. Can more advanced methods being applied to this work? like PPO as it also has a A2C network.

---

### Official Review · AnonReviewer2 · 2020-12-19
**Some suggestions for this paper.**

**Rating:** 7
**Confidence:** 4

**Review:**

In this paper, the authors mainly study the problem that enhances the automatic generation of code summarization by developing a   RL framework. They adopt the Actor-critic network to solve the problem and their results show that their model performs well, which outperforms vallina sequence to sequence model by a noticeable margin.

Here are some suggestions for your paper and I hope it could help you further improve your paper:

1.It's better to write a few key words in the abstract, which will help readers to quickly understand the core of your paper;

2.In your experiments, hope you can explain the experiment settings in detail, such as the settings of the hyper parameters and network structure. In addition, when describing the dataset, maybe you can give examples to illustrate the composition of your dataset more clearly;

3.For the results, it converges after about 1000 steps, so the later training may not very useful. Maybe you can improve the performance of your model by adjusting the learning rate;

4.There seem to be some typos in the paper. For example, in the third paragraph of section 1, I think it should be "In this paper" but not "proposal"? Maybe you can check it again.

Finally thank you for letting me know another use of RL.